# Nitroglycerin for treatment of retained placenta: A randomised, placebo-controlled, multicentre, double-blind trial in the UK

**Fiona C. Denison**[1]*, **Kathryn F. Carruthers**[1], **Jemma Hudson**[2], **Gladys McPherson**[2], **Gin Nie Chua**[3], **Mathilde Peace**[1], **Jane Brewin**[4], **Nina Hallowell**[5,6], **Graham Scotland**[2,3], **Julia Lawton**[7], **John Norrie**[8], **Jane E. Norman**[1], GOT-IT investigator team

**1** Tommy's Centre for Maternal and Fetal Health, Medical Research Council Centre for Reproductive Health, Queen's Medical Research Institute, University of Edinburgh, Edinburgh, United Kingdom, **2** Centre for Healthcare Randomised Trials, Health Services Research Unit, University of Aberdeen, Aberdeen, United Kingdom, **3** Health Economics Research Unit, Institute of Applied Health Sciences, University of Aberdeen, Aberdeen, United Kingdom, **4** Tommy's, London, United Kingdom, **5** Wellcome Centre for Ethics and Humanities, Nuffield Department of Population Health, University of Oxford, Oxford, United Kingdom, **6** Ethox Centre, Nuffield Department of Population Health, University of Oxford, Oxford, United Kingdom, **7** Usher Institute of Population Health Sciences and Informatics, University of Edinburgh, Edinburgh, United Kingdom, **8** Edinburgh Clinical Trials Unit, Usher Institute of Population Health Sciences and Informatics, University of Edinburgh, Edinburgh, United Kingdom

\* Fiona.Denison@ed.ac.uk

**Data Availability Statement:** The GOT-IT Trial contains a centrally-managed cross centre dataset which is available, upon request, from the Digital

## Abstract

### Background

Retained placenta following vaginal delivery is a major cause of postpartum haemorrhage. Currently, the only effective treatments for a retained placenta are the surgical procedures of manual removal of placenta (MROP) and uterine curettage, which are not universally available, particularly in low- and middle-income countries. The objective of the trial was to determine whether sublingual nitroglycerin spray was clinically effective and cost-effective for medical treatment of retained placenta following vaginal delivery.

### Methods and findings

A randomised, placebo-controlled, double-blind trial was undertaken between October 2014 and July 2017 at 29 delivery units in the UK (Edinburgh, Glasgow, Manchester, Newcastle, Preston, Warrington, Chesterfield, Crewe, Durham, West Middlesex, Aylesbury, Furness, Southampton, Bolton, Sunderland, Oxford, Nottingham [2 units], Burnley, Chertsey, Stockton-on-Tees, Middlesborough, Chester, Darlington, York, Reading, Milton Keynes, Telford, Frimley). In total, 1,107 women with retained placenta following vaginal delivery were recruited. The intervention was self-administered 2 puffs of sublingual nitroglycerin (800 μg; intervention, $N = 543$) or placebo spray (control, $N = 564$). The primary clinical outcome was the need for MROP, assessed at 15 minutes following administration of the intervention. Analysis was based on the intention-to-treat principle. The primary safety outcome was measured blood loss between study drug administration and transfer to the postnatal ward or other clinical area. The primary patient-sided outcomes were satisfaction with treatment

Curation Centre within the University of Edinburgh. Data requests can be made at crh-data-manager@ed.ac.uk.

**Funding:** The GOT-IT trial was funded by the UK National Institute for Health Research (NIHR; https://www.nihr.ac.uk) Health Technology Assessment (HTA) Program in response to a specific commissioned grant call (Project number 12/29/01). The following co-authors were grant holders: FCD, GM, MP, JB, GS, JL, JN and JEN. The funders played no role in the study design, data collection, analysis, decision to publish or preparation of the manuscript. This work was undertaken in the MRC Centre for Reproductive Health which is funded by MRC Centre grant (MRC G1002033).

**Competing interests:** I have read the journal's policy and the authors of this manuscript have the following competing interests: FCD is named as Principal Investigator on government and charitable research grants to their institution which aim to improve pregnancy outcome. JEN is named as Principal Investigator on government and charitable research grants to their institution which aim to improve pregnancy outcome. In the last three years JEN has provided consultancy to pharma companies GSK and Dilafor; my institution was renumerated for this. JEN's institution has received travel and subsistence expenses from Merck to facilitate them speaking at a Merck sponsored symposium on metformin. JEN is on Subpanel A1 for REF, and on a Wellcome Trust Science interview panel, and receive personal renumeration for each. KFC is named as CoInvestigator on a charitable research grant about coronary syndrome. None of the other co-authors have declared any competing interests.

**Abbreviations:** bpm, beats per minute; iDMC, independent data monitoring committee; ITT, intention-to-treat; MROP, manual removal of placenta; OR, odds ratio; TSC, trial steering committee.

and side-effect profile, assessed by questionnaires pre-discharge and 6 weeks post-delivery. Secondary clinical outcomes were measured at 5 and 15 minutes after study drug administration and prior to hospital discharge. There was no statistically significant or clinically meaningful difference in need for MROP by 15 minutes (primary clinical outcome, 505 [93.3%] for nitroglycerin versus 518 [92.0%] for placebo, odds ratio [OR] 1.01 [95% CI 0.98–1.04], $p = 0.393$) or blood loss (<500 ml: nitroglycerin, 238 [44.3%], versus placebo, 249 [44.5%]; 500 ml–1,000 ml: nitroglycerin, 180 [33.5%], versus placebo, 224 [40.0%]; >1,000 ml: nitroglycerin, 119 [22.2%], versus placebo, 87 [15.5%]; ordinal OR 1.14 [95% CI 0.88–1.48], $p = 0.314$) or satisfaction with treatment (nitroglycerin, 288 [75.4%], versus placebo, 303 [78.1%]; OR 0.87 [95% CI 0.62–1.22], $p = 0.411$) or health service costs (mean difference [£] 55.3 [95% CI −199.20 to 309.79]). Palpitations following drug administration were reported more often in the nitroglycerin group (36 [9.8%] versus 15 [4.0%], OR 2.60 [95% CI 1.40–4.84], $p = 0.003$). There were 52 serious adverse events during the trial, with no statistically significant difference in likelihood between groups (nitroglycerin, 27 [5.0%], versus placebo, 26 [4.6%]; OR 1.13 [95% CI 0.54–2.38], $p = 0.747$). The main limitation of our study was the low return rate for the 6-week postnatal questionnaire. There were, however, no differences in questionnaire return rates between study groups or between women who did and did not have MROP, with the patient-reported use of outpatient and primary care services at 6 weeks accounting for only a small proportion (approximately 5%) of overall health service costs.

## Conclusions

In this study, we found that nitroglycerin is neither clinically effective nor cost-effective as a medical treatment for retained placenta, and has increased side effects, suggesting it should not be used. Further research is required to identify an effective medical treatment for retained placenta to reduce the morbidity caused by this condition, particularly in low- and middle-income countries where surgical management is not available.

## Trial registration

ISRCTN.com ISRCTN88609453
ClinicalTrials.gov NCT02085213

## Author summary

### Why was this study done?

- A retained placenta can cause life-threatening bleeding in women following a vaginal birth.
- The only effective treatment for a retained placenta is for it to be removed by an operation.
- In many parts of the world, surgery is not possible, meaning that women die from this condition.

## What did the researchers do and find?

- We undertook a trial to assess whether a drug (nitroglycerin) to relax the womb would be an effective, safe, and acceptable medical treatment for retained placenta and would avoid the need for surgical removal.

- We recruited 1,107 women with retained placenta and randomised them to treatment with sublingual nitroglycerin or placebo spray to treat retained placenta.

- We found that nitroglycerin was not effective as a medical treatment for retained placenta.

## What do these findings mean?

- Our findings indicate that sublingual nitroglycerin does not effectively reduce the need for women with a retained placenta following vaginal delivery to have the placenta removed by an operation.

- These findings indicate that there remains a need for a new, safe, and effective medical treatment for retained placenta for those women who live in settings where operative treatment for retained placenta is not available.

## Introduction

Retained placenta following childbirth complicates 0.1%–2% of deliveries [1]. Without prompt treatment, it results in significant haemorrhage, which can result in maternal death. Current treatment for retained placenta is the surgical procedure of manual removal of placenta (MROP) or uterine curettage, which has attendant risks including bleeding and infection. This procedure is not available in all settings, particularly in low- and middle-income countries, where retained placenta has a high morbidity and mortality rate [2,3]. There is therefore a need for an effective, acceptable, safe, and affordable medical treatment for retained placenta that is suitable for all settings.

Small studies have suggested that nitric oxide donors such as nitroglycerin (also known as glyceryl trinitrate) may be an effective treatment for retained placenta [4]. Six studies (5 case series [5–9] and 1 small placebo-controlled randomised trial [10]) have reported that administration of nitroglycerin intravenously [5,6,8,9] or via a sublingual tablet [7,10] is effective in relaxing the uterus to facilitate insertion of the examining hand for MROP [5,6,8,9] or in facilitating delivery of the placenta by controlled cord traction [7,10]. However, these findings were not replicated in 2 other studies in which nitroglycerin (intravenous [11] or sublingual tablet [12]) was not effective in medical treatment of retained placenta.

If nitroglycerin is to be effective for medical management of retained placenta, it must be able to address at least 1 of the underlying pathophysiological mechanisms. In placentae that are detached but trapped behind a myometrial constriction ring, nitroglycerin could potentially relax local uterine muscle constriction, thereby effecting placental release. In adherent placenta, Farley et al. have suggested that nitric-oxide-mediated contraction and relaxation of human chorionic villi along their longitudinal axis might serve as a nitroglycerin-mediated

mechanism for placental separation [13]. Where the placenta is morbidly adherent to the myometrium, currently available nitric oxide donor drugs (including nitroglycerin) are unlikely to effect release, and surgical management is likely to remain the mainstay of treatment. In summary, although there are signals that nitroglycerin may have potential to medically treat retained placenta, there is a need to undertake a high-quality randomised, placebo-controlled, multicentre, double-blind trial to definitively determine whether nitroglycerin is or is not effective in medically managing retained placenta [5–9,14–17].

The GOT-IT (Glyceryl Trinitrate for Retained Placenta) trial was a large multi-centre trial that aimed to determine the clinical effectiveness and cost-effectiveness of sublingual nitroglycerin (glyceryl trinitrate) spray compared with placebo in reducing the need for MROP in women with retained placenta after vaginal delivery.

## Methods

### Study design and oversight

The GOT-IT trial was funded by the UK National Institute for Health Research Health Technology Assessment Programme in response to a specific commissioned grant call. Details of the trial protocol (S1 Text) and statistical analysis plan (S2 Text) have been published previously [18]. The North East–Newcastle and North Tyneside 2 Research Ethics Committee (13/NE/0339) approved the trial. A trial steering committee (TSC) and independent data monitoring committee (iDMC) provided trial oversight (S3 Text). Information about the trial was made available to women during pregnancy. Clinical staff approached eligible women with retained placenta, and, following discussion, informed written consent—or oral consent followed up by written consent as soon as possible—was obtained from women who were interested in taking part in the trial. Further details about the consent and recruitment processes and the steps taken to ensure that participants gave truly informed consent are provided in the trial protocol (S1 Text). The authors vouch for the accuracy and completeness of the data and for the fidelity of the trial to the protocol. This study is reported as per the Consolidated Standards of Reporting Trials (CONSORT) guideline (S1 Checklist). The trial was registered in the ISRCTN registry (http://www.isrctn.com/, ISCRTN88609453).

### Trial setting and patients

From 1 October 2014 to 31 July 2017, women diagnosed with retained placenta following vaginal delivery were identified and screened for eligibility by clinical staff in delivery wards in 29 maternity hospitals in England and Scotland, UK (S1 Table). Maternity units were selected based on their ability to undertake an intrapartum research study, and included both teaching hospitals and district general hospitals, with delivery numbers ranging from approximately 1,000 per annum to >7,000 per annum.

Women were eligible for the trial if, following vaginal birth, they sustained a retained placenta and were at risk of needing MROP. A retained placenta was defined according to National Institute for Health and Care Excellence guidelines as the placenta remaining undelivered after 30 minutes of active management of the third stage of labour [19].

Women were eligible if they delivered at >14 weeks gestation, were ≥16 years of age, and were haemodynamically stable (defined as heart rate ≤ 119 beats per minute [bpm] and systolic blood pressure > 100 mm Hg).

We excluded women with suspected placenta accreta/increta/percreta; allergy, hypersensitivity, or contraindication to nitrates; alcohol consumption within the past 24 hours; instrumental vaginal delivery in an operating theatre; multiple pregnancy in the index pregnancy; or inability to give informed consent or who were taking phosphodiesterase inhibitor.

Women were given information about the trial antenatally. Eligible women with retained placenta were approached by clinical staff, and, following discussion, informed written consent—or oral consent followed up by written consent as soon as possible—was obtained from women who indicated willingness to take part in the trial. Further details about these processes are provided in the trial protocol (S1 Text) [18] and qualitative research publications [20–22]. Baseline demographics, including maternal age, body mass index (BMI), smoking status, ethnicity, and alcohol use, were obtained from the antenatal booking record by the local research teams (Table 1).

## Randomisation and masking

We randomly assigned participants (1:1) to nitroglycerin or placebo. Study medication was provided in pre-packed randomised permuted blocks and stored in temperature-controlled storage areas in delivery rooms. Study drug and placebo were manufactured by Pharmasol and labelled by Sharp Clinical Services (UK). Once a participant was recruited, the study drug was

**Table 1. Baseline characteristics of study participants.**

| Characteristic | Nitroglycerin (*N* = 541) | Placebo (*N* = 563) |
|---|---|---|
| Age (years)—mean (SD); *n* | 30.6 (5.5); 541 | 30.8 (5.1); 563 |
| BMI (kg/m$^2$)—mean (SD); *n* | 25.8 (5.4); 526 | 25.4 (5.2); 548 |
| Smoker | | |
| Current smoker | 75 (13.9) | 77 (13.7) |
| Ex-smoker | 101 (18.7) | 98 (17.4) |
| Never smoker | 350 (64.7) | 376 (66.8) |
| Missing | 15 (2.8) | 12 (2.1) |
| Alcohol use in pregnancy | | |
| Yes | 19 (3.5) | 18 (3.2) |
| No | 505 (93.3) | 521 (92.5) |
| Missing | 17 (3.1) | 24 (4.3) |
| Ethnicity | | |
| White | 468 (86.5) | 487 (86.5) |
| Asian | 38 (7.0) | 41 (7.3) |
| Black | 7 (1.3) | 8 (1.4) |
| Mixed | 5 (0.9) | 6 (1.1) |
| Chinese | 5 (0.9) | 6 (1.1) |
| Other | 5 (0.9) | 6 (1.1) |
| Missing | 13 (2.4) | 9 (1.6) |
| Blood pressure (mm Hg)—mean (SD); *n* | | |
| Systolic | 123.8 (12.8); 538 | 124.6 (12.6); 559 |
| Diastolic | 73.3 (10.2); 535 | 75.1 (10.1); 559 |
| Heart rate (bpm)—mean (SD); *n* | 84.6 (13.0); 539 | 84.7 (12.9); 559 |
| Temperature (˚C)—mean (SD); *n* | 36.8 (0.5); 513 | 36.9 (0.4); 534 |
| Haemoglobin (mmol/l)—mean (SD); *n* | 7.6 (0.8); 468 | 7.6 (0.9); 478 |
| Previous pregnancy | 311 (57.5) | 323 (57.4) |
| Previous retained placenta | 48 (15.4) | 57 (17.6) |
| Previous placenta praevia/accreta | 4 (1.3) | 1 (0.3) |

Values are *n* (%) unless otherwise stated.

bpm, beats per minute.

allocated by taking the next available treatment pack from the shelf. Both clinicians and participants were blinded to the treatment allocation.

### Trial interventions

Baseline observations (maternal blood pressure [mm Hg] and heart rate [bpm]) were taken prior to study drug administration. If eligibility was confirmed, women self-administered 2 puffs of sublingual nitroglycerin (800 µg; intervention) or placebo spray (control). Maternal temperature (˚C) and haemoglobin were measured at baseline, with heart rate, systolic blood pressure, and temperature being recorded at 5 and 15 minutes after study drug administration. If the placenta remained undelivered at 15 minutes after study drug administration, the decision was made to proceed with MROP as soon as possible.

A haemoglobin sample was collected on the first postnatal day. Questionnaires were completed prior to hospital discharge (patient satisfaction and side effects) and at 6 weeks post-delivery (patient satisfaction, side effects, and health resource use) (S4 and S5 Texts).

### Trial outcomes

The primary clinical outcome was the need for MROP, i.e., the placenta remaining undelivered 15 minutes after study treatment and/or MROP being required within 15 minutes of treatment due to safety concerns. The primary safety outcome was measured blood loss between study drug administration and transfer to the postnatal ward or other clinical area. Blood loss was measured using the routine clinical methods used at study sites, and was categorised as <500 ml, 500–1,000 ml, or >1,000 ml by local investigators. The primary patient-sided outcomes were satisfaction with treatment and side-effect profile, assessed by questionnaires pre-discharge (satisfaction and treatment-associated side effects) and at 6 weeks post-delivery (satisfaction, side effects in the 6 weeks following delivery, and health resource use). The primary economic outcome was a comparison of the use of nitroglycerin versus standard practice by evaluating the net incremental costs to the UK National Health Service.

Secondary clinical outcomes were fall in haemoglobin of >15% between randomisation and the first postnatal day; time from randomisation to delivery of placenta; MROP in theatre; need for earlier than planned MROP due to clinical condition; fall in systolic or diastolic blood pressure of >15 mm Hg and/or increase in heart rate of >20 bpm between baseline and 5 and 15 minutes after administration of treatment; need for blood transfusion between time of delivery and discharge from hospital; need for general anaesthesia; maternal pyrexia (1 or more temperature readings of >38˚C prior to discharge from hospital, or in the first 72 hours after delivery, if hospital stay was longer than 72 hours); and sustained uterine relaxation after removal of placenta requiring uterotonics.

### Safety, adverse event monitoring, and trial management

All reported adverse events were documented in the participants' clinical records and collated and coded by the trial office, which was located in the Queen's Medical Research Institute at the University of Edinburgh. The trial office comprised the trial manager and trial administrator. The trial office was responsible for the day-to-day running of the trial and worked closely with the Centre for Healthcare Randomised Trials at the University of Aberdeen. The chief investigator (FCD) had regular oversight of the trial office.

All serious adverse events were reported by the principal site investigator to the sponsor and the trial office, and also entered into the electronic database within 24 hours of the site becoming aware of the event. The chief investigator was notified of all severe adverse event reports, and all events were followed up until resolution. The TSC and iDMC reviewed the

serious adverse events at regular meetings every 6 months, with the latter reviewing the data unblinded. If any serious concerns had arisen about trial safety, the chair of the iDMC would have recommended to the chair of the TSC that the study should be discontinued. Finally, our lay advisors were involved throughout study, with their advice influencing the trial design and delivery, and informing project management group and TSC discussions.

## Statistical analyses

From discussions with clinicians and women, we determined that an absolute benefit (i.e., reduction in the need for MROP) of 10% would be the minimum required to make it worth implementing the intervention (nitroglycerin spray) in practice. We therefore took a statistical approach to the control (placebo) rate, setting this at 50% because, from a statistical perspective, this corresponds to the highest variability in a yes/no (binary) outcome, and hence generates the largest maximum sample size required to demonstrate an absolute 10% difference. That meant that we could be confident that if the observed control rate was higher or lower, our study would be sufficiently powered to detect a 10% absolute difference and potentially adequately powered to detect smaller absolute differences.

Due to considerable uncertainties in the untreated rate and expected treatment effect, we adopted a group sequential design with 5 interim analyses, allowing the iDMC the flexibility to end the study if there was overwhelming evidence of benefit or futility. Allowing for 5 interim analyses, 90% power, and 5% significance level, a maximum sample size of 1,078 participants was needed to demonstrate a 10% change in rate, from 50% on placebo to 40% on nitroglycerin spray. We used a Lan–DeMets alpha spending approach [23] with O'Brien–Fleming boundaries [24]. We specified a 2-sided test, with efficacy and futility boundaries (only for the third interim analysis) and 5 interim reviews by the iDMC, equally spaced at 215, 429, 644, 858, and 1,073 randomised participants with primary outcome data. Analysis was based on the intention-to-treat (ITT) principle. Statistical significance was at the 2-sided 5% level, with the working level of significance set at $p = 0.0481$.

The group sequential design for the primary clinical outcome was analysed using logistic regression with no adjustment for centre, using East 6.4.1 (2016) [25]. The primary safety outcome was analysed using ordered logistic regression, and patient-sided outcomes were analysed using logistic regression, accounting for centre using cluster robust standard errors. Secondary clinical outcomes were analysed using logistic or linear regression, as appropriate. Continuous variables were summarised with mean and standard deviation, and discrete variables were summarised as absolute number and percentage. The remainder of the analysis was undertaken using Stata 14 [26]. Pre-specified subgroup analyses (previous cesarean section and gestation at delivery <36 and ≥36 weeks) were conducted using a stricter 2-sided 1% level of statistical significance. A post hoc analysis looked at serious adverse events and was conducted using logistic regression adjusting for centre.

A cost analysis was undertaken to quantify the difference in mean costs between the nitroglycerin and placebo arm. Research costs associated with placebo delivery were factored out of the analysis to estimate the incremental cost (or cost savings) of the active intervention versus standard practice. Resource use associated with the alternative management strategies was estimated from the time of randomisation to 6 weeks postpartum. Resources included staff time for providing study drug to patients, additional resource use associated with complications arising following administration of study drug, subsequent costs associated with delivery of the placenta, costs associated with postnatal stay (to discharge), and costs associated with subsequent health service contact relating to retained products of conception up to 6 weeks post-delivery. National unit price data were used to attach costs to the different elements of resource

use (S2 Table) [27–31]. Mean costs were summarised by treatment allocation group, and the incremental cost (or cost savings) associated with the use of nitroglycerin was estimated using linear regression with cluster robust standard errors.

## Results

### Patients

From October 2014 through July 2017, 1,671 women were screened, 564 were excluded, and 1,107 (66%) women were subsequently consented and randomly assigned, 564 to placebo and 543 to nitroglycerin (Fig 1).

Of the 1,671 women screened, 353 were ineligible, 63 declined, 60 were missed, and it was not appropriate to approach 7 patients (S3 Table). A further 81 eligible women were excluded prior to randomisation, and 3 participants were excluded post-randomisation (2 in the nitroglycerin arm and 1 in the placebo arm), making the ITT analysis set 541 in the nitroglycerin arm versus 563 in the placebo arm. Twelve participants did not receive the study drug (6 in the nitroglycerin arm and 7 in the placebo arm) (S4 Table). In total, 390 participants in the nitroglycerin arm and 399 in the placebo arm filled in the pre-discharge questionnaire. At the 6-week follow-up, 228 participants in the nitroglycerin arm and 241 in the placebo arm returned the questionnaire. The baseline characteristics of participants in the ITT population were balanced across the 2 allocated groups (Table 1).

### Primary outcomes

The trial was not stopped early at any of the interim analysis stages, as none of the stopping boundaries were crossed. These boundaries were generated assuming a control rate of 50%. In practice, the control rate was never less than 90%. The TSC and iDMC (which had seen unblinded data) had a series of discussions about re-estimating the sample size and changing the timings of the interim analyses, but the unanimous decision was to carry on unaltered. The rationale was that there did not seem to be any emerging safety concerns, recruitment was going well, and therefore the original sample size should be maintained to (1) get precise estimates of any treatment effect (or lack of it) and (2) have as much power as possible for secondary outcomes and the economic evaluation, particularly given that more participants than expected were progressing on blinded data to MROP (S1 Fig). The trial therefore recruited to its maximum size of 1,104 randomised.

There was no statistically significant or clinically meaningful difference in the primary clinical outcome (need for MROP by 15 minutes) between the 2 groups (505 [93.3%] for nitroglycerin versus 518 [92.0%] for placebo, odds ratio [OR] 1.01 [95% CI 0.98 to 1.04], $p = 0.39$) (Table 2). For participants where the placenta was delivered after 15 minutes, the majority had MROP (nitroglycerin, 407 [80.6%]; placebo, 417 [80.5%]; unadjusted OR 1.00 [95% CI 0.83–1.20], $p = 0.97$) (S5 Table).

There was no statistically significant or clinically meaningful difference in the primary safety outcome of blood loss between groups (<500 ml: nitroglycerin, 238 [44.3%], versus placebo, 249 [44.5%]; 500–1,000 ml: nitroglycerin, 180 [33.5%], versus placebo, 224 [40.0%]; >1,000 ml: nitroglycerin, 119 [22.2%], versus placebo, 87 [15.5%]; ordinal OR 1.14 [95% CI 0.88–1.48], $p = 0.31$) (Table 2).

There was no statistically significant or clinically meaningful difference in the primary patient-sided satisfaction outcome of recommending the study drug to a friend/relative at pre-discharge (nitroglycerin, 288 [75.4%], versus placebo, 303 [78.1%]; OR 0.87 [95% CI 0.62–1.22], $p = 0.41$) or at 6 weeks (nitroglycerin, 166 [75.1%], versus placebo, 178 [74.8%]; OR 1.02 [95% CI 0.66–1.56], $p = 0.94$). For the primary patient-sided side-effect outcome, participants

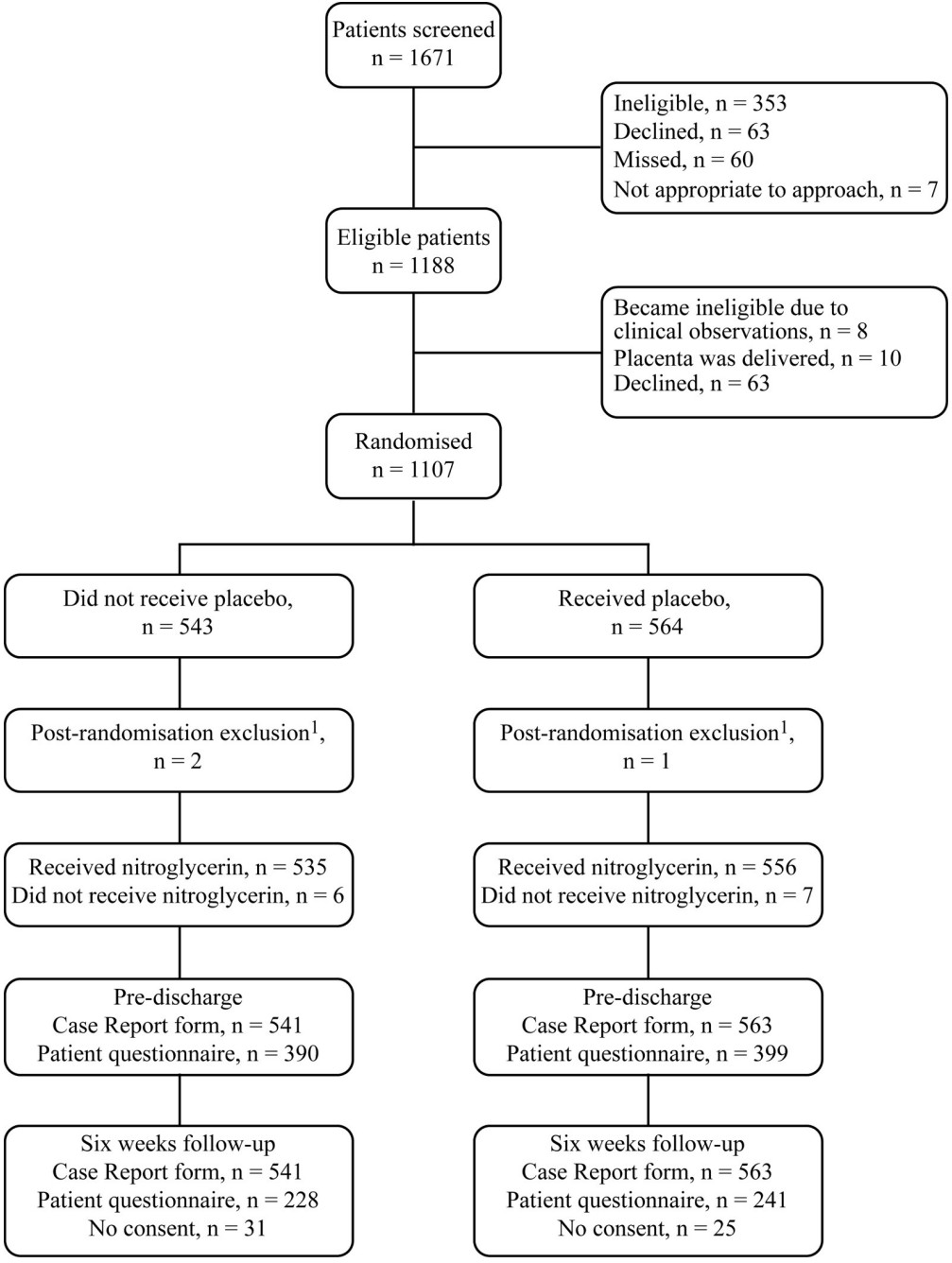

**Fig 1. CONSORT diagram for GOT-IT trial.** [1]Reasons for post-randomisation exclusions were (1) eligibility could not be confirmed as the last observations were taken approximately 1 hour prior to the study drug being given; (2) the inclusion criteria observations were not documented; and (3) the participant consent form was lost and there were no eligible observations documented.

who received nitroglycerin were more likely to report palpitations/heart racing prior to discharge (nitroglycerin, 36 [9.8%], versus placebo, 15 [4.0%]; OR 2.60 [95% CI 1.40–4.84], $p = 0.003$). There was no statistically significant difference in palpitations/heart racing by 6

**Table 2. Primary clinical, safety, and patient-sided outcomes.**

| Outcome | Nitroglycerin (N = 541) | Placebo (N = 563) | Odds ratio (95% CI) | p-Value |
|---|---|---|---|---|
| **Primary clinical outcome**[*] | | | | |
| Placenta delivered within 15 minutes | 36/541 (6.7) | 45/563 (8.0) | | |
| Placenta not delivered within 15 minutes | 505/541 (93.3) | 518/563 (92.0) | 1.01 (0.98, 1.04) | 0.39 |
| **Primary safety outcome: blood loss** | | | | |
| <500 ml | 238/537 (44.3) | 249/560 (44.5) | | |
| 500–1,000 ml | 180/537 (33.5) | 224/560 (40.0) | | |
| >1,000 ml | 119/537 (22.2) | 87/560 (15.5) | 1.14 (0.88, 1.48) | 0.31 |
| **Primary patient-sided outcomes** | | | | |
| Recommend study drug to a friend/relative? | | | | |
| **Pre-discharge** | | | | |
| No | 94/382 (24.6) | 85/388 (21.9) | | |
| Yes | 288/382 (75.4) | 303/388 (78.1) | 0.87 ((0.62, 1.22) | 0.41 |
| **6 weeks** | | | | |
| No | 55/221 (24.9) | 60/238 (25.2) | | |
| Yes | 166/221 (75.1) | 178/238 (74.8) | 1.02 (0.66, 1.56) | 0.94 |
| Feeling sick | | | | |
| **Pre-discharge** | | | | |
| No | 299/377 (79.3) | 323/384 (84.1) | | |
| Yes | 78/377 (20.7) | 61/384 (15.9) | 1.37 (0.94, 1.99) | 0.10 |
| **6 weeks** | | | | |
| No | 180/211 (85.3) | 206/232 (88.8) | | |
| Yes | 31/211 (14.7) | 26/232 (11.2) | 1.40 (0.80, 2.47) | 0.23 |
| Palpitations/heart racing | | | | |
| **Pre-discharge** | | | | |
| No | 332/368 (90.2) | 360/375 (96.0) | | |
| Yes | 36/368 (9.8) | 15/375 (4.0) | 2.60 (1.40, 4.84) | 0.003 |
| **6 weeks** | | | | |
| No | 186/200 (93.0) | 215/225 (95.6) | | |
| Yes | 14/200 (7.0) | 10/225 (4.4) | 1.62 (0.70, 3.73) | 0.25 |

Values are n/N (%).

[*]This analysis was adjusted for multiple looks at the data in accordance with the group sequential design.

weeks postnatal (nitroglycerin, 31 [14.7%], versus placebo, 26 [11.2%]; OR 1.40 [95% CI 0.80–2.47]; p = 0.24) and no statistically significant difference in feeling sick pre-discharge or at 6 weeks between the 2 groups (Table 2).

Health service resource use per patient is summarised by treatment allocation group in S6 Table. Table 3 summarises the associated mean health service costs by treatment allocation, along with the estimated difference between groups. There were no statistically significant differences between the groups in any of the cost categories, although hospital episode costs were non-statistically-significantly higher in the nitroglycerin arm, which was driven by a slightly higher MROP rate.

## Secondary clinical outcomes

Secondary clinical outcomes are shown in Table 4. Participants in the nitroglycerin group were more likely than those in the placebo group to have a fall in systolic or diastolic blood pressure and/or increase in heart rate between baseline and 5 and 15 minutes post-

**Table 3. Difference in NHS per-patient costs by category and treatment allocation (intention to treat).**

| Category | Number of observations[a] | Cost (£), mean (SD) (N = 1,104) | | Mean difference in cost (£) (95% CI)[b] |
|---|---|---|---|---|
| | | Nitroglycerin | Placebo | |
| Total episode cost | 966 | 1,366.62 (733.61) | 1,317.12 (642.42) | 49.50 (−42.63 to 141.64) |
| Total primary care cost | 424 | 25.13 (52.29) | 28.40 (58.59) | −3.28 (−13.93 to 7.38) |
| Cost of outpatient appointment | 466 | 25.65 (98.42) | 18.86 (63.96) | 6.79 (−10.79 to 24.37) |
| Cost of hospital readmission | 1,098 | 52.05 (858.84) | 43.32 (263.98) | 8.73 (−61.92 to 79.39) |
| Total NHS cost[c] | 369 | 1,513.95 (1,732) | 1,458.65 (779) | 55.30 (−199.20 to 309.79) |

[a]Number of observations with complete data on each cost category.

[b]Cluster robust CIs.

[c]Incorporates total episode cost, total primary care cost, cost of outpatient appointments, and cost of hospital readmissions in individuals with complete data across all categories.

NHS, National Health Service.

**Table 4. Secondary clinical outcomes.**

| Outcome | Nitroglycerin (N = 541) | Placebo (N = 563) | Effect size[a] (95% CI) | p-Value |
|---|---|---|---|---|
| **Fall in systolic or diastolic blood pressure and/or increase in heart rate[b]** | | | | |
| No | 208/531 (39.2) | 413/544 (75.9) | | |
| Yes | 323/531 (60.8) | 131/544 (24.1) | 4.90 (3.73, 6.42) | <0.001 |
| **Blood transfusion** | | | | |
| No | 472/533 (88.6) | 508/551 (92.2) | | |
| Yes | 61/533 (11.4) | 43/551 (7.8) | 1.53 (1.04, 2.25) | 0.03 |
| **More than 15% fall in haemoglobin** | | | | |
| No | 160/414 (38.6) | 180/421 (42.8) | | |
| Yes | 254/414 (61.4) | 241/421 (57.2) | 1.19 (0.93, 1.52) | 0.18 |
| **Time from randomisation to delivery of placenta (mins)** | | | | |
| | 12.1 (7.3); 539 | 12.2 (7.0); 561 | −0.19 (−0.94, 0.55) | 0.60 |
| **Manual removal of placenta in theatre** | | | | |
| No | 141/540 (26.1) | 152/563 (27.0) | | |
| Yes | 399/540 (73.9) | 411/563 (73.0) | 1.05 (0.80, 1.36) | 0.74 |
| **Need for earlier than planned manual removal of placenta** | | | | |
| No | 407/416 (97.8) | 420/431 (97.4) | | |
| Yes | 9/416 (2.2) | 11/431 (2.6) | 0.84 (0.30, 2.35) | 0.75 |
| **General anaesthesia** | | | | |
| No | 390/438 (89.0) | 398/443 (89.8) | | |
| Yes | 48/438 (11.0) | 45/443 (10.2) | 1.09 (0.66, 1.80) | 0.74 |
| **Maternal pyrexia** | | | | |
| No | 516/527 (97.9) | 530/551 (96.2) | | |
| Yes | 11/527 (2.1) | 21/551 (3.8) | 0.54 (0.26, 1.11) | 0.09 |
| **Sustained uterine relaxation[c]** | | | | |
| No | 460/528 (87.1) | 482/550 (87.6) | | |
| Yes | 68/528 (12.9) | 68/550 (12.4) | 1.05 (0.76, 1.44) | 0.77 |

Values are n/N (%) for dichotomous variables and mean (SD); n for continuous variables.

[a]Effect sizes are odds ratios apart from time from randomisation to delivery of placenta, which is mean difference.

[b]Defined as fall in systolic or diastolic blood pressure of more than 15 mm Hg and/or increase in heart rate of more than 20 beats per minute between baseline and 5 and 15 minutes post-administration.

[c]Defined as uterine relaxation requiring additional uterotonics.

administration (nitroglycerin, 323 [60.8%], versus placebo, 131 [24.1%]; OR 4.90 [95% CI 3.73–6.42], $p < 0.001$) and to require a blood transfusion between time of delivery and discharge from hospital (nitroglycerin, 61 [11.4%], versus placebo, 43 [7.8%]; OR 1.53 [95% CI 1.04–2.25], $p = 0.03$). There were no statistically significant or clinically meaningful differences between the groups for any other secondary clinical outcomes. As there was a low event rate for the primary clinical outcome, the number of events in the subgroups was too low to perform the planned subgroup analysis.

### Safety outcomes

There were 52 serious adverse events during the trial (nitroglycerin, 27 [5.0%]; placebo, 26 [4.6%]). The majority required hospitalisation (nitroglycerin, 24; placebo, 26) and were due to postpartum haemorrhage (nitroglycerin, 17; placebo, 12) (S7 Table).

## Discussion

To our knowledge, GOT-IT is the largest multi-centre randomised, trial of nitroglycerin for medical treatment of retained placenta in women following vaginal delivery, and was powered to detect an absolute 10% difference in efficacy, assuming an untreated rate of requiring MROP of 50%. In contrast to previous publications suggesting that nitric oxide donors may be an effective treatment for retained placenta [4,5–9,10], our larger and more robust trial demonstrates that nitroglycerin is ineffective for medical treatment of retained placenta when used with controlled cord traction. There were no statistically significant or clinically meaningful differences in the primary clinical, safety, patient-sided, or economic outcomes, with the observed non-statistically-significant differences in effectiveness and safety outcomes directionally favouring placebo. Secondary clinical outcomes also suggested increased side-effect profile and haemodynamic changes following nitroglycerin administration, with the increased number of blood transfusions signalling possible safety concerns.

We recruited over 1,100 women with an obstetric emergency to a clinical trial of a medicinal product in an acute peripartum setting from a large number of centres of differing size, increasing the generalisability of the results. We achieved our recruitment target ahead of time, and within budget. A key strength of our trial is our flexible group sequential trial design, which enabled us to accommodate uncertainties in the key evidence on which this trial was based, and included the ability to stop the trial if there was evidence of overwhelming efficacy or futility. As none of the efficacy and futility boundaries were crossed at any of the interim analyses, the trial proceeded to recruit to its full sample size, albeit with a dramatically higher event rate than expected (>90% compared with the assumed 50%). This allowed, in the presence of much lower binomial variability, very precise estimates of the lack of a treatment effect, enabling us to confidently rule out any meaningful clinical benefit from this intervention.

A potential weakness in our trial was the low return rate for the 6-week postnatal questionnaire. Although the return rate improved following implementation of recommendations from qualitative research [21,22], the overall rate remained disappointingly low. We attributed this poor return rate to women having insufficient time to complete and return a questionnaire while caring for a newborn child. There were, however, no differences in return rates between study groups or between women who did and did not have MROP, and the patient-reported use of outpatient and primary care services at 6 weeks accounted for only a small proportion (approximately 5%) of overall health service costs.

Nitroglycerin can be administered either sublingually or intravenously. Both routes of administration have the same pharmacokinetic properties, with the onset of action within 2 to 3 minutes and peak plasma concentrations approximately 6 to 7 minutes post-dose [32]. We

chose to use sublingual spray because, compared to sublingual tablets, the spray has several advantages including significant reduction in latency of onset, fewer objective and subjective side effects, and stability at room temperature [33]. Nitroglycerin spray is also used in other obstetric emergencies where rapid uterine relaxation is required, for example to release a trapped head in cesarean section or breech delivery. We discounted the intravenous route of administration because, although intravenous nitroglycerin has been used to treat retained placenta, the requirement for cannulation limits its generalisability, and the symptomatic hypotension that occurs at higher doses would be potentially dangerous in low- and middle-income settings where options for resuscitation are limited. Although it is possible that our results may have been different if we had used a different route of administration, we believe this is unlikely. Women who received the nitroglycerin were more likely to report palpitations and to have a fall in systolic or diastolic blood pressure and/or increase in heart rate following its administration. This is consistent with the known effects of nitroglycerin and provides evidence that self-administration of the intervention by women was effective in causing a pharmacological effect.

One of the challenges of treating retained placenta is that its pathophysiology remains poorly understood. A retained placenta is a clinical diagnosis that is reported to variously occur when the placenta is detached but trapped, or partially or completely adherent to the underlying myometrium. Although ultrasound has been used in a research setting to try to phenotype retained placentae, its diagnostic accuracy and utility to inform clinical management is not proven. Ultrasound is also not readily available in low- and middle-income settings. We therefore chose a pragmatic approach to trial inclusion, with women being eligible for trial entry if they had a clinical diagnosis of retained placenta. We accept that if different phenotypes of retained placenta do exist that respond differently to different treatments, we may not have been able to identify this in our trial. This may have contributed to our finding that nitroglycerin was ineffective for management of retained placenta.

To inform our trial design, we consulted our lay advisors and clinicians about whether a woman who had had an instrumental vaginal delivery in an operating theatre should be eligible to take part in the trial. Our consultees expressed concerns that it would be undignified and unethical for a woman to remain exposed in a theatre environment whilst waiting for a retained placenta to be diagnosed. They felt that these concerns were less when the instrumental delivery occurred in the delivery room because this was a more private space, where it would easier for those involved in the delivery to maintain the mother's dignity. Given this strong steer, we decided that women with an instrumental vaginal delivery in theatre would not be eligible to take part in the study. The main difference between women having an instrumental vaginal delivery in a delivery room (who were eligible) and in theatre is the setting: We therefore believe that our finding, that nitroglycerin is ineffective for treatment of retained placenta, is generalisable to women with instrumental vaginal delivery in theatre.

A potential safety concern was that nitroglycerin-induced uterine relaxation might increase blood loss. Although there was no evidence that blood loss was higher with nitroglycerin, women randomised to nitroglycerin were more likely to receive a blood transfusion. Given that there was no significant difference in drop in haemoglobin between groups (Table 4), we are unclear why administration of nitroglycerin was associated with an increased transfusion rate. However, we speculate that the haemodynamic changes caused by nitroglycerin administration might have altered clinician behaviour in favour of transfusion.

In conclusion, our trial indicates that sublingual nitroglycerin spray is neither clinically effective nor cost-effective for medical treatment of retained placenta when used with controlled cord traction following vaginal delivery and should not be used for this indication. Of note, among women whose placenta remained undelivered 15 minutes after administration of

the study drug, over 80% in both study groups were required to go to theatre for MROP. There therefore remains a need for an effective, acceptable, safe, and affordable medical treatment for retained placenta, particularly for low- and middle-income countries, where MROP in theatre is often not available.

## Supporting information

**S1 Checklist. CONSORT checklist.**
(DOC)

**S1 Fig. Stopping boundaries.**
(DOCX)

**S1 Table. Recruitment by centre.**
(DOCX)

**S2 Table. Identification, measurement, and valuation of resource use.**
(DOCX)

**S3 Table. Description of participants who were excluded pre-randomisation.**
(DOCX)

**S4 Table. Reasons why study drug was not given.**
(DOCX)

**S5 Table. Method of placenta removal.**
(DOCX)

**S6 Table. Resource use by treatment allocation (intention to treat).**
(DOCX)

**S7 Table. Severe adverse events.**
(DOCX)

**S1 Text. GOT-IT trial protocol.**
(PDF)

**S2 Text. Statistical analysis plan.**
(PDF)

**S3 Text. Trial oversight committees: trial steering committee and independent data monitoring committee.**
(DOCX)

**S4 Text. GOT-IT trial pre-discharge questionnaire.**
(DOCX)

**S5 Text. GOT-IT trial 6-week postnatal questionnaire.**
(DOCX)

## Acknowledgments

This work was undertaken in the MRC Centre for Reproductive Health, which is funded by an MRC centre grant (MRC G1002033). The views and opinions expressed therein are those of the authors and do not necessarily reflect those of the Health Technology Assessment

Programme, the National Institute for Health Research, the National Health Service, or the Department of Health.

## Author Contributions

**Conceptualization:** Fiona C. Denison, Julia Lawton, John Norrie.

**Data curation:** Fiona C. Denison, Kathryn F. Carruthers, Jemma Hudson.

**Formal analysis:** Fiona C. Denison, Jemma Hudson, Gin Nie Chua, Nina Hallowell, Graham Scotland, John Norrie, Jane E. Norman.

**Funding acquisition:** Fiona C. Denison, Gladys McPherson, Mathilde Peace, Jane Brewin, Graham Scotland, Julia Lawton, John Norrie, Jane E. Norman.

**Investigation:** Fiona C. Denison, Gladys McPherson, Gin Nie Chua, Mathilde Peace, Jane Brewin, Nina Hallowell, Graham Scotland, Julia Lawton, John Norrie, Jane E. Norman.

**Methodology:** Fiona C. Denison, Jemma Hudson, Gladys McPherson, Nina Hallowell, Graham Scotland, Julia Lawton, John Norrie, Jane E. Norman.

**Project administration:** Fiona C. Denison, Kathryn F. Carruthers, Gladys McPherson, Graham Scotland, Julia Lawton, John Norrie.

**Supervision:** Fiona C. Denison, Graham Scotland, Julia Lawton, John Norrie.

**Writing – original draft:** Fiona C. Denison, Kathryn F. Carruthers, Graham Scotland, Julia Lawton, John Norrie, Jane E. Norman.

**Writing – review & editing:** Fiona C. Denison, Kathryn F. Carruthers, Jemma Hudson, Gladys McPherson, Gin Nie Chua, Mathilde Peace, Jane Brewin, Nina Hallowell, Graham Scotland, Julia Lawton, John Norrie, Jane E. Norman.

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
