## [Decision Letter · Decision Letter 0]

20 Sep 2019

Dear Dr. Denison,

Thank you very much for submitting your manuscript "Glyceryl trinitrate for treatment of retained placenta: a randomized, placebo-controlled, multicentre double blind trial" (PMEDICINE-D-19-02549) for consideration at PLOS Medicine. 

Your paper was evaluated by a senior editor and discussed among all the editors here. It was also discussed with an academic editor with relevant expertise, and sent to three independent reviewers, including a statistical reviewer. The reviews are appended at the bottom of this email and any accompanying reviewer attachments can be seen via the link below:

[LINK]

In light of these reviews, I am afraid that we will not be able to accept the manuscript for publication in the journal in its current form, but we would like to consider a revised version that addresses the reviewers' and editors' comments. Obviously we cannot make any decision about publication until we have seen the revised manuscript and your response, and we plan to seek re-review by one or more of the reviewers. 

We expect to receive your revised manuscript by Oct 04 2019 11:59PM. Please email us (plosmedicine@plos.org) if you have any questions or concerns.

We look forward to receiving your revised manuscript. 

Sincerely,

Caitlin Moyer, Ph.D.

Associate Editor 

PLOS Medicine

plosmedicine.org

1. The following outcomes measures pertaining to the economic outcomes appear to differ between the submitted manuscript and the trial registry. Please clarify and explain the discrepancy. If the outcomes were not prespecified in the protocol, please indicate that they were post hoc and explain why they were added. Post hoc comparisons should be presented as hypothesis generating rather than conclusive. 

Specifically, a primary economic outcome of comparing GTN vs standard practice in terms of net incremental cost to UK NHS (page 7, Lines 174-176). However, the following primary and secondary outcomes included in the trial registry are slightly different (http://www.isrctn.com/ISRCTN88609453): Primary outcome: 4. Economic: net incremental costs (or cost savings) to the National Health Service of using GTN versus standard practice. Costs will include GTN (dose and time to administer drug, monitor woman and deliver the placenta if effective), MROP, and further health service resource use as measured by the 6-week postnatal questionnaire; Secondary outcome: 2. Costs: the mean costs will be summarised by treatment allocation group, and the incremental cost (cost saving) associated with the use of GTN will be estimated using an appropriately specified general linear model. The cost data will be presented alongside the primary and secondary outcome data in a cost-consequence balance sheet, indicating which strategy each outcome favours (6-week postnatal questionnaire). 

2. Title: Please make it clear in the title that the trial took place in the UK. 

3. Abstract Line 44: October 14 should be October 2014. 

4. Abstract Line 43-44: The second instance of “was undertaken” is a typo- please correct. Also, “29 delivery units” should read “at 29 delivery units”.

5. Abstract Results: Please quantify the main results (with 95% CIs and p values) for the adverse outcomes presented (e.g. palpitations, adverse events).

6. Abstract methods and results: Please indicate the outcomes assessed at follow up, and length and method of follow up.

7. Abstract methods and results: Please indicate the locations (i.e. in which cities) of the maternity wards involved in the study.

9. Methods and Results: Please provide further detail regarding the size and other relevant characteristics of the 29 maternity wards included in the study, including how the hospitals were chosen.

10. Methods and Results: The questionnaire administered at hospital discharge and 6 week follow up (mentioned at lines 161-163) should be included as a supporting document. Please include a file containing the questionnaire, and provide a reference to it in the text. 

11. Line 188: Please clarify the identity of the Chief Investigator.

12. Line 188: Please clarify the identity, location, and the role of the “Trial Office”.

13. Line 225: Please remove the discussion of the cost analysis. Because the results of the trial were negative it is not necessary.

14. Line 239: Please remove the section “Role of the Funding Source” from the main text. This is not needed as the information will be automatically extracted from the manuscript submission data.

15. Methods and Results: Please be consistent with numbers of decimal places when reporting p values. In general, use two decimal places for p>.01 and three decimal places for p<.01 (e.g. Table 2 vs. Table 4).

16. Discussion: Please qualify the first sentence of the discussion with the phrase “to our knowledge” or similar. (Line 339: “GOT-IT is the largest multi-centre, randomised, trial...”) 

17. Discussion, Line 384: Please clarify what is meant by, and the role of the “lay advisors” mentioned. 

18. Discussion: Please elaborate more on where and why the results may differ from previous research/studies of nitroglycerin treatment for retained placenta. 

19. Throughout the manuscript: Please use square brackets “[]” for in-text reference numbers, rather than parentheses. Formatting guidelines can be found at: https://journals.plos.org/plosmedicine/s/submission-guidelines#loc-references

20. For Supplementary Figures 1 and 2, please specify in the X axis label that ‘Sample Size’ denotes number of participants.

21. CONSORT checklist: Thank you for including the CONSORT checklist. However, some of the page numbers are inconsistent, please double check and update throughout (e.g. Eligibility criteria for participants seem to be described on page 6, rather than page 7-10 as indicated).

Comments from the reviewers:

Reviewer #1: This is, by far, the largest reported trial of glyceryl trinitrate (GTN) for treatment of retained placenta (RP) and the only trial of sub-lingual spray. The methodological quality of the trial is strong. I don't understand some of the statistical methods, but recognise that these will be reviewed by a statistical referee. The trial clearly needs to be published somewhere prominent, where it will be noticed.

I do, however, think that the reporting of the trial could in some respects be improved. 

The relevant Cochrane review by Abdel-Aleem and colleagues (2015) - strangely uncited in the paper (it should be) - describes 3 randomised trials of GTN for RP (2 sub-lingual tablets, 1 intravenous); the combined incidence of manual removal of placenta (MROP) in the control groups was 80% and even though two of the trials had slightly more conservative time criteria for intervention (40 and 50 minutes) than GOT-IT, it is difficult to understand why the trial team planned on the basis of 50% MROP in the control group. This mis-match is only addressed in the paper in relation to the decision-making of the data monitoring committee in light of the unexpected incidence of the primary outcome. This should be discussed more fully.

On a related topic, retained placentas are thought to either be detached but retained in the uterus (and potentially amenable to delivery after GTN relaxes uterine muscle) or non-detached (and much less likely to be aided by GTN). I don't know if there are good data on the incidence of each. My clinical impression is that the latter are much more common. I think these pathophysiological considerations need to be discussed in more depth to clarify the rationale for the trial, and to refine the discussion about sample size calculation.

The rationale for the treatment regimen is not discussed in much detail other than saying that this was what the funder wanted for this commissioned research call. There are presumably data on bio-equivalence with other GTN regimens from eg internal medicine research. Why this treatment at this dosage? What evidence prior to trial commencement that this is efficacious? Current discussion could be expanded.

There are several references to the importance of RP in low and middle income countries. The authors are correct to emphasise this. They might wish to expand a little on where they saw GTN having a possible role (should it have proven effective). Manual removal of placenta is one of the functions of the Basic Emergency Obstetric Care package, but access to health centres can often be problem. Did they see a potential role for medical intervention for RP in community care or in health centres or both?

Surely the trial entry criterion 'delivery > 14 weeks' is a mistake (line 131)?

Jim Neilson

Reviewer #2: This manuscript reports the results of a RCT comparing Glyceryl trinitrate to placebo for the treatment of retained placenta. The manuscript is well written and very easy to follow. I only have minor comments listed below:

* In the method section, please indicate the nominal type-I error rate available at the end of the trial given the overall 5% level and the 5 interim reviews planned

* East 6.4.1 was used to analyse the primary clinical outcome. Please clarify the method (model, adjustment) used for the primary clinical outcome. In particular, please clarify whether the model adjust for the effect of centre?

* Please confirm that all patients with a CRF available post-discharge also had a primary outcome (MROP use) available

* The footnote in Table 2 states that "the analysis was adjusted for multiple looks at the data". Please clarify what this means. E.g. where the confidence intervals adjusted? the p-values? Is this true for all analyses presented in Table 2 or only the primary clinical outcome?

* The discussion states (lines 356-357) that "none of the efficacy and futility boundaries were crossed at any of the interim analysis" (a similar statement is made in lines 269-270); however, according to supplementary Figure 2, the futility boundaries were crossed at the 4th interim analysis. Please clarify/correct as well as explain more clearly this apparent inconsistency. 

-Laurent Billot

Reviewer #3: This is a very well-written and well-desisgned trial. The clarity in writing and transparency in reporting is commendable. 

I have only minor recommendations for this manuscript, which are as follows: 

1) In the title or abstract, please include the term "nitroglycerin," to ensure broader understanding of glyceryl-trinitrate (GTN) as "nitroglycerin" is the more common term used in the U.S. and other countries. Consider changing the abbreviation "GTN" as this is commonly used to represent "gestational trophoblastic neoplasia"

2) Lines 38-39 and 79-80: the authors discuss that the only alternative management for retained placenta is manual removal, and do not discuss curettage, a commonly performed procedure for retained placenta. This, too, may not be available in lower resource settings, but deserves mention

3) Lines 95-96: The authors correctly mention that glyceryl trinitrate may be effective for retained placenta in the presence of a contraction ring. The authors appear to have used the medication as a solo measure. A limitation of this study that is not mentioned, but that is more clinically relevant is how often nitroglycerin was needed to allow a contraction ring to relax sufficiently to allow passage of the providers hand to complete manual removal of the placenta, followed by uterotonics and therefore, reduce blood loss. With this in mind, I recommend that the authors consider modifying their conclusion to state that glyceryl trinitrate is neither clinically effective nor cost effective when used alone, and either provide evidence that it is ineffective in this very clinically relevant subset or mention that it may still be considered in this setting, absent other recourse.

[LINK]

---

## [Decision Letter · Decision Letter 1]

1 Nov 2019

Dear Dr. Denison,

Thank you very much for re-submitting your manuscript "Glyceryl trinitrate for treatment of retained placenta: a randomized, placebo-controlled, multicentre double blind trial" (PMEDICINE-D-19-02549R1) for review by PLOS Medicine.

I have discussed the paper with my colleagues and the academic editor and it was also seen again by one reviewer. I am pleased to say that provided the remaining editorial and production issues are dealt with we are planning to accept the paper for publication in the journal.

[LINK]

We look forward to receiving the revised manuscript by Nov 08 2019 11:59PM. 

Sincerely,

Caitlin Moyer, Ph.D.

Associate Editor 

PLOS Medicine

plosmedicine.org

Requests from Editors:

1.Response to reviewer comments: Thank you for your response to Reviewer 1, point 2 (R1, C2) concerning your rationale for assuming a 50% rate of MROP in the control group. Please also clarify this rationale in the manuscript, also, either in the Statistical Analyses section of the Methods or elsewhere as appropriate.

2.Data Availability Statement: Thank you for noting that data cannot be shared publicly, and providing a link for requesting access to data. However, this link connects with a very general web page. If possible, please provide a more specific link and/or a contact email address to facilitate access to data.

3.Abstract: Background: Please define the abbreviation (RP) at first use.

4.Abstract: Line 68: “There were however no difference in questionnaire return rates…”: Please revise to “There were however no differences in questionnaire return rates…”.

5.Abstract: Methods and Findings: Please state that analysis was intention to treat.

6.Abstract: Methods and Findings: Please provide the number in each group.

7.Abstract: Conclusions: Please address the study implications without overreaching what can be concluded from the data; the phrase "In this study, we observed ..." may be useful. Please interpret the study based on the results presented in the abstract, emphasizing what is new without overstating your conclusions. Specifically, please revise the first sentence to: “In this study, we found that nitroglycerin is neither clinically-effective nor cost-effective as a medical treatment for retained placenta, and has increased side effects, suggesting it should not be used.” or similar.

8. Author Summary: “Why was this study done?”: Please revise the first point to clarify what is meant by “clinically important” bleeding, for example: “A retained placenta can cause life-threatening bleeding in women following a vaginal birth” or similar.

9. Author Summary: “What do these findings mean?”: Please clarify the first bullet point to: “Our findings indicate that sublingual nitroglycerin does not effectively reduce the need for women with a retained placenta following vaginal delivery to have the placenta removed by an operation.” or similar.

10. Author Summary: “What do these findings mean?”: Please revise the second bullet point to: “These findings indicate that there remains a need for a new, safe, and effective medical treatment for retained placenta for those women who live in settings where operative treatment for retained placenta is not available.” or similar.

11. Introduction: Line 105-108: Please provide some reference(s) for the sentence describing how surgical MROP is not available in LMIC.

12. Methods: Line 153: Please change “hospital” to “hospitals”.

13. Methods: Line 165: Please revise to “We excluded...placenta accreta/increta/percreta; allergy, hypersensitivity, or contra-indication to nitrates; alcohol…”.

14. Methods: Line 223: Please clarify to “...collated and coded by the Trial Office, which was located in the Queen’s Medical Research Institute…” or similar.

15. Methods: Line 227: Please clarify to: “The Chief Investigator (Fiona Denison) had regular oversight of the trial office.” or similar.

16. Results: Line 314: “Placebo” should not need capitalization.

17. Results: Line 318-319: Please indicate which values correspond to nitroglycerin vs. placebo groups for the blood loss outcomes.

18. Results: Line 325-326: Please indicate which values correspond to nitroglycerin vs. placebo groups for the patient satisfaction outcomes.

19. Results: Line 328 and 330: Please indicate which values correspond to nitroglycerin vs. placebo groups for the palpitations/heart racing outcomes.

20. Results: Line 348 and 350: Please indicate which values correspond to nitroglycerin vs. placebo groups for the blood pressure and blood transfusion outcomes.

21. Discussion: Line 454: Please clarify the use of the term “significant” used in this context. If statistical significance is indicated, please consider revising to: “Given that there was no significant difference in drop in haemoglobin between groups (Table 4), we are unclear…”.

22. Discussion: Line 406-408: Thank you for your response to editor request C18. However, Please provide references and expand on how your findings specifically are in contrast to the previous publication findings (i.e. the sentence beginning “In contrast to previous publications…”).

23. Discussion: Line 382: Please remove the sentence “Our trial was highly successful.” 

24. Discussion: Line 388-389: Please revise to “As none of the efficacy and futility boundaries were crossed at any of the interim analyses, the trial was allowed to recruit to its full sample size…” or similar to clarify the meaning.

25. Discussion: Line 413-414: Please revise to “...the requirement for cannulation limits its generalisability…”

26. Discussion: Lines 424-425 (and throughout): Please consistently use either “retained placenta” or the abbreviation “RP” throughout the manuscript, defining the abbreviation at first use.

27. Discussion: Line 438: Thank you for clarifying the consulting role of the lay advisors in response to editor request C17. Please mention somewhere in the manuscript why and how lay advisors were involved with the study.

28. Table 1: Please define the abbreviations for “bpm” and “SD” in the table footnote.

29. Table 2: Please define the abbreviation for “CI” in the table footnote. Please describe in the table footnote the adjustment for multiple looks at the data.

30. Table 3: Please define the abbreviations for “CI” and “SD” in the table footnote.

31. Table 4: Please define the abbreviation for “CI” in the table footnote.

32. Supplementary Table 5: PLease remove the trademark symbol for “Coro Nitro” and indicate in parentheses after that this is the glyceryl trinitrate. 

33. Supplementary Table 7: Please define the abbreviation “bpm” in the table footnote.

34. Checklist: Thank you for providing the CONSORT checklist. At this time please replace the page numbers with paragraph numbers per section (e.g. "Methods, paragraph 1"), as the page numbers of the final published paper may be different from the page numbers in the current manuscript.

Comments from Reviewers:

Reviewer #2: My comments have been adequately adressed.

-Laurent Billot

[LINK]

---

## [Editor Report · Decision Letter 2]

22 Nov 2019

Dear Prof. Denison, 

On behalf of my colleagues and the academic editor, Dr. Jenny Myers, I am delighted to inform you that your manuscript entitled "Glyceryl trinitrate for treatment of retained placenta: a randomized, placebo-controlled, multicentre double blind trial" (PMEDICINE-D-19-02549R2) has been accepted for publication in PLOS Medicine. The manuscript is scheduled to publish on December 30, 2019. 

PRODUCTION PROCESS

PRESS

PROFILE INFORMATION

Thank you again for submitting the manuscript to PLOS Medicine. We look forward to publishing it. 

Best wishes, 

Caitlin Moyer, Ph.D.

Associate Editor 

PLOS Medicine

plosmedicine.org